# Systemic Redox Imbalance in Patients with Chronic Granulomatous Disease

**DOI:** 10.3390/jcm9051397

**Published:** 2020-05-09

**Authors:** Edyta Heropolitanska-Pliszka, Klaudia Berk, Mateusz Maciejczyk, Jolanta Sawicka-Powierza, Ewa Bernatowska, Beata Wolska-Kusnierz, Malgorzata Pac, Nel Dabrowska-Leonik, Barbara Piatosa, Aleksandra Lewandowicz-Uszynska, Joanna Karpinska, Anna Zalewska, Bozena Mikoluc

**Affiliations:** 1Clinical Immunology the Children’s Memorial Health Institute, al. Dzieci Polskich 20, 04-730 Warsaw, Poland; ehp@poczta.onet.eu (E.H.-P.); ewa.bernatowska@gmail.com (E.B.); bwolska@interia.pl (B.W.-K.); malgorzata.pac@wp.pl (M.P.); nel.dabrowska@wp.pl (N.D.-L.); 2Department of Physiology, Medical University of Bialystok, ul. Mickiewicza 2c, 15-233 Bialystok, Poland; klaudia.berk@gmail.com; 3Department of Hygiene, Epidemiology and Ergonomics, Medical University of Bialystok, ul. Mickiewicza 2c, 15-233 Bialystok, Poland; mat.maciejczyk@gmail.com; 4Department of Family Medicine, Medical University of Bialystok, 15-054 Bialystok, Poland; jolasawicka@gmail.com; 5Histocompatibility Laboratory, Children’s Memorial Health Institute, al. Dzieci Polskich 20, 04-730 Warsaw, Poland; B.Piatosa@IPCZD.PL; 63rd Department and Clinic of Pediatrics, Immunology and Rheumatology of Developmental Age, Wroclaw Medical University, ul. Koszarowa 5, 50-367 Wrocław, Poland; aleksandra.lewandowicz-uszynska@umed.wroc.pl; 7Institute of Chemistry, University of Bialystok, ul. Ciołkowskiego. 1K, 15-245 Białystok, Poland; joasia@uwb.edu.pl; 8Experimental Dentistry Laboratory, Medical University of Bialystok, ul. Szpitalna 37, 15-295 Bialystok, Poland; azalewska426@gmail.com; 9Department of Pediatrics, Rheumatology, Immunology and Metabolic Bone Diseases, Waszyngtona 17, 15-274 Bialystok, Poland

**Keywords:** chronic granulomatous disease, primary immunodeficiency, oxidative stress, antioxidants, coenzyme Q10

## Abstract

The aim of our study was to evaluate redox status, enzymatic and non-enzymatic antioxidant barriers, oxidative damage of proteins, lipids and DNA, as well as concentration of coenzyme Q10 and vitamins A and E in patients with chronic granulomatous disease (CGD). The study was performed on fifteen Caucasian individuals (median age 24 years and seven months) diagnosed with CGD. The mutation in the *NCF1* gene was confirmed in ten patients, and in the *CYBB* gene in five patients. We demonstrated high levels of total oxidant status (TOS) and oxidative stress index (OSI), lipids (↑8-isoprostanes (8-isoP), ↑4-hydroxynonenal (4-HNE)), proteins (↑advanced oxidation protein products (AOPP)) and DNA (↑8-hydroxy-2’-deoxyguanosine (8-OHdG)) oxidation products in CGD individuals as compared to sex- and age-matched healthy controls. We showed enhanced serum enzymatic activity of catalase (CAT) and superoxide dismutase-1 (SOD) and significantly decreased coenzyme Q10 concentration. Our study confirmed redox disturbances and increased oxidative damage in CGD patients, and indicated the need to compare redox imbalance depending on the type of mutation and nicotinamide adenine dinucleotide phosphate (NADPH) oxidase activity. The question regarding effectiveness of antioxidant therapy in patients with CGD is open, and the need to establish guidelines in this area remains to be addressed.

## 1. Introduction

Chronic granulomatous disease (CGD) is a rare (one in 200,000–250,000 live births), hereditary, primary immunodeficiency disorder (PID) [1]. The hallmark of CGD is impairment in superoxide production caused by mutation of one of six genes that encode proteins in nicotinamide adenine dinucleotide phosphate (NADPH) oxidase complex (NOX2) present mainly in professional phagocytes [2]. NADPH oxidase transfers electrons from cytosol to molecular oxygen across the phagosomal membrane, leading to the formation of superoxide [3]. In principle, the largest group of patients (approximately 60–70%) is composed of males because of mutation in the *CYBB* gene encoding gp91phox in the X chromosome. Other gene mutations, including *NCF1* (gp47phox), *NCF2* (gp67phox), *NCF4* (gp40phox), *CYBA* (gp22phox) and *CYBC1* (also known as EROS) are inherited in autosomal recessive fashion [2].

Patients with CGD demonstrate chronic or recurrent severe bacterial (catalase-positive) and fungal infections, predominantly *Aspergillus* spp. [4]. Other symptoms include a variety of inflammatory conditions unrelated to infections such as granuloma formation, colitis and increased frequency of autoimmune diseases [5]. Data on neoplasm development in CGD are limited despite the fact that PIDs in general are susceptible to malignancy [6]. Although the complexity regarding the role of ROS in cancer is significant but divergent, a study by van der Weyden et al. indicates that NOX2 significantly influences the process of metastasis in mice genetically deprived of any of the major NOX2 subunits, and consistently shows reduced lung metastasis after intravenous injection of tumor cells [7]. Several studies confirm that CGD can be regarded as an autoinflammatory disease [8]. Recently, increased reactive oxygen species (ROS)-independent inflammasome activation has been shown in NOX2-deficient phagocytes [9]. It has been suggested that increased production of mitochondrial ROS in CGD phagocytes together with other immune abnormalities are key contributors to the hyperinflammatory features of CGD [10]. Considering available literature data and the results of our study, we present possible mechanisms of dysregulated inflammatory response in the course of CGD (Figure 1). Despite the proinflammatory nature of immune cells previously reported in CGD patients, redox homeostasis has not yet been evaluated in this group of patients. Therefore, the aim of our study was to assess redox status, enzymatic and non-enzymatic antioxidant barriers, oxidative damage to proteins, lipids and DNA, as well as levels of coenzyme Q10 and vitamins A and E in patients with CGD in comparison to healthy controls.

## 2. Material and Methods

### 2.1. Study Population

Fifteen Caucasian patients with CGD (ten males and five females, median age 24.7 ± 13.7 years) were recruited for this study.

Diagnosis was established in accordance with European Society for Immunodeficiencies (ESID) criteria [11]. Mutations in the *NCF1* gene were confirmed in ten patients, and mutations in the *CYBB* gene were confirmed in five patients. All patients included in the study were found to be in good health and were receiving itraconazole, trimethoprim and sulfamethoxazole at the time of enrollment. No side effects of the drugs used were observed. Apart from CGD, none of the patients demonstrated any other pathologies or exclusion criteria, i.e., metabolic diseases (obesity, diabetes), hypertension, heart, liver, kidney or lung diseases (cystitis fibrosis, allergic bronchopulmonary aspergillosis), HIV infection, cancer, hyper IgE syndrome, glutathione synthetase or myeloperoxidase deficiency and secondary hemophagocytic lymphohistiocytosis. Based on histopathological examination, six patients with CGD showed mild inflammatory bowel disease (IBD), but without clinical symptoms and not requiring treatment. In all patients with CGD, markers of inflammation (C-reactive protein, CRP; leukocytosis; erythrocyte sedimentation rate, ESR) were negative. 

The control group consisted of 15 healthy age- and sex-matched individuals. All study and control subjects had the same dietary habits, and none took vitamins or antioxidant supplements.

All patients participating in the study were under the medical care of the Department of Immunology at the Children’s Memorial Health Institute in Warsaw, Poland.

The study was approved by the local Ethics Committee of the Medical University of Bialystok, Poland (No: R-I-002/300/2018). All patients or legal guardians of patients below 16 years of age provided written informed consent for participation in the project.

### 2.2. Blood Samples

Following an overnight fast, venous blood (5 mL) was collected into tubes containing ethylenediaminetetraacetic acid (S-Monovette^®^ EDTA-K3, Sarstedt, Germany) and into tubes with clot activator (S-Monovette^®^ Clotting Activator/Serum, Sarstedt, Germany). In accordance with the manufacturer’s instructions, blood samples were protected from light, centrifuged (2000× *g* for 10 min at +4 °C), and then plasma and serum were immediately separated. Blood was frozen at a temperature of −80 °C until assayed, for no longer than 6 months.

### 2.3. Redox Status

Plasma redox status parameters: total antioxidant status (TAS), total oxidant status (TOS) and oxidative stress index (OSI) were determined using commercial colorimetric kits (ImAnOx (TAS/TAC) Kit, Immundiagnostik, Bensheim, Germany and PerOx (TOS/TOC) Kit, Immundiagnostik, Bensheim, Germany, respectively) in accordance with the manufacturer’s instructions. Determination of TAS was based on the reaction between antioxidants contained in the sample with exogenous hydrogen peroxide, while determination of TOS was based on the reaction of peroxidase with total lipid peroxides in the sample. The resulting colored products were measured colorimetrically at 450 nm. Oxidative stress index (OSI) was calculated according to the following formula: OSI = TOS/TAS × 100% [12].

### 2.4. Redox Assays

The performed analyses included antioxidant enzymes: glutathione peroxidase (GPx, E.C. 1.11.1.9), catalase (CAT, E.C. 1.11.1.6) and superoxide dismutase-1 (SOD, E.C. 1.15.1.1); non-enzymatic antioxidants (uric acid (UA) and oxidative damage products): advanced glycation end products (AGE), advanced oxidation protein products (AOPP), 4-hydroxynonenal (4-HNE) protein adducts, 8-isoprostanes (8-isoP) and 8-hydroxy-2’-deoxyguanosine (8-OHdG). Absorbance/fluorescence was measured using the Infinite M200 PRO Multimode Microplate Reader (Tecan Group Ltd., Männedorf, Switzerland). All determinations were performed in duplicate samples and standardized for 100 mg of total protein. 

a. Antioxidant barrier: serum GPx activity was measured using a method based on the conversion of NADPH to NADP^+^ described by Paglia et al. [13]. Absorbance was analyzed at 340 nm. One unit of GPx activity was defined as the amount of enzyme catalyzing the oxidation of 1 mmol NADPH for 1 min [13]. 

Serum CAT activity was determined spectrophotometrically by measuring hydrogen peroxide (H_2_O_2_) decomposition at 340 nm. One unit of CAT activity was defined as the amount of enzyme which degrades 1 μmol of H_2_O_2_ for 1 min. CAT activity was determined in triplicate samples [14]. Serum SOD activity was evaluated by measurement of the inhibition products of epinephrine to adrenochrome oxidation. Absorbance was analyzed at 340 nm. It was assumed that one unit of SOD activity inhibits oxidation of epinephrine by 50%. SOD activity was determined in triplicate samples [15]. 

Plasma UA concentrations were determined spectrophotometrically using a commercial kit (QuantiChrom™ Uric Acid Assay Kit, DIUA-250, BioAssay Systems, Harward, CA, USA) in accordance with the manufacturer’s instructions. Absorbance was measured at 490 nm. 

Analyses of plasma lipophilic antioxidants (retinol, α-tocopherol, coenzyme Q10) were performed using a high performance liquid chromatograph coupled to an MS detector equipped with a triple quadrupole (Shimadzu LCMS/MS-8040). Ionization was conducted using the atmospheric pressure chemical ionization (APCI) mode. Data acquisition and processing were performed using Shimadzu LabSolutions LCMS software. The compounds were separated with a Kinetex XB-C18 100A analytical column (50 mm × 3.0 mm, 1.7 μm). The mobile phase consisted of an isocratic solvent A (methanol) for 0.01–2 min and then isocratic solvent B (methanol-n-hexan, 72:28, *v*/*v*) for 2.5–6 min. Flow rate was 0.4 mL/min and temperature of the analytical column was 400 °C. Injection volumes of standard and sample solutions were 10 μL. Acquisition settings and method were optimized by the infusion of a 10 μg/mL solution of each fixed compound. The mass spectrometer was operated in the positive ion atmospheric pressure chemical ionization mode. APCI temperature was set at 350 °C and ion current at 4.5 μA. The flow rate of the drying gas (N_2_) and nebulizing gas were 10 L/min and 3 L/min, respectively. The desolvation line (DL) and heat block temperature was 230 °C. All analytes were detected in MS/MS multiple reaction monitoring (MRM) with unit resolution at both Q1 and Q3. MS conditions for generating positive ions are presented in Table 1. The chromatographic conditions described above were used for the quantification of target analytes in plasma samples. Plasma samples were prepared according to the procedure published previously [16].

b. Oxidative damage products of proteins, lipids and DNA. Plasma AGE content was determined spectrofluorimetrically using the method of Kalousová et al. [17]. Fluorescence of plasma samples was measured at excitation wavelength 440 nm and emission wavelength 350 nm. For AOPP determination, plasma samples were diluted 1:5 (*v*:*v*) in phosphate-buffered saline (PBS) (pH 7.2) [17,18].

Plasma AOPP concentrations were analyzed spectrophotometrically by measuring the oxidative capacity of the iodine ion at 340 nm. For AOPP determination, serum samples were diluted 1:5 (*v*:*v*) in PBS (pH 7.2) [17,18].

Plasma 4-HNE, 8-isoP and 8-OHdG concentrations were estimated using a commercial enzyme-linked immunosorbent assay (ELISA) in accordance with the manufacturer’s instructions (Cell Biolabs, Inc., San Diego, CA, USA; Cayman Chemicals, Ann Arbor, MI, USA; USCN Life Science, Wuhan, China, respectively).

### 2.5. Statistical Analysis

Statistical analysis was performed using parametric tests for normally distributed variables (Student’s *t*-test and Pearson’s correlation method). The data were processed using Statistica 12.0 (StatSoft, Cracow, Poland) and GraphPad Prism 7 (GraphPad Software, La Jolla, CA, USA) and expressed as mean ± SD and median. Statistical significance was defined as *p* ≤ 0.05.

## 3. Results

Basic clinical data of CGD patients are presented in Table 2.

Our study revealed that in patients with CGD, mean levels of plasma TOS (11.26 ± 3.14 vs. 6.8 ± 3.3 µmol/L, *p* < 0.0001) and OSI (1.53 ± 0.51 vs. 1.0 ± 0.6, *p* < 0.05) were statistically higher than in the control group. Plasma levels of TAS did not differ between CGD patients and controls (7.82 ± 3.62 vs. 7.93 ± 2.78 µmol/L, *p* = 0.93) (Figure 2).

We demonstrated significantly enhanced activity of antioxidant enzymes such as CAT (5.99 ± 1.88 vs. 3.3 ± 1.46 nmol/H_2_O_2_/mg of protein, *p* < 0.0001) and SOD (1.09 ± 0.41 vs. 0.74 ± 0.26 mU/mg of protein, *p* < 0.05) in the sera of patients with CGD as compared to the control group. However, we did not observe significant differences in GPx serum activity between patients with CGD and controls (0.1 ± 0.01 vs. 0.11 ± 0.03 mU/mg protein). Levels of UA were similar in CGD patients and healthy controls (0.73 ± 0.22 vs. 0.56 ± 0.35 µg/mg of protein *(p* = 0.12)) (Figure 3).

Among all evaluated endogenous free radical scavengers, we noticed significantly decreased plasma coenzyme Q10 concentration in CGD patients as compared to the control group (0.06 ± 0.05 vs. 0.31 ± 0.16 μg/mL, *p* < 0.0001). Both tocopherol (1.76 ± 0.94 vs. 1.63 ± 0.79 μg/mL, *p* = 0.67) and retinol (0.09 ± 0.07 vs. 0.07 ± 0.03 μg/mL, *p* = 0.29) plasma concentrations did not differ between the two groups, patients and controls (Figure 2).

We demonstrated oxidative damage to DNA (↑8-OHdG, 38.41 ± 16.26 vs. 21.27 ± 16.26 pg/mg of protein, *p* = 0.0029), proteins (↑AOPP, 4.43 ± 1.72 vs. 2.6 ± 1.6 nmol/mg protein, *p* < 0.0001) and lipids (↑4-HNE, 0.18 ± 0.07 vs. 0.08 ± 0.03 µg/mg protein, *p* = 0.0005 and ↑8-isoP, 0.83 ± 0.17 vs. 0.56 ± 0.14 pg/mg protein, *p* < 0.0001)). In contrast, plasma AGE concentrations in CGD patients and healthy controls were similar (445.6 ± 152.3 vs. 370.6 ± 127.4 fluorescence/mg protein, *p* = 0.15) (Figure 4).

In Table 3, we present a comparison of selected redox parameters between patients with the *CYBB* and the *NCF1* mutation. There were no differences in the assessed parameters of redox status between patients with the *CYBB* and the *NCF1* mutation.

NADPH activity in patients with chronic granulomatous disease is measured as percentage of granulocytes demonstrating change in dihydrorhodamine fluorescence after stimulation with phorbol myristate, presented in Figure 5.

## 4. Discussion

Literature data regarding clinical symptoms occurring in CGD suggest a critical role of oxidative stress in the pathogenesis of this syndrome. To our knowledge, this is the first study to evaluate selected biomarkers of redox homeostasis in CGD patients and reveal systemic redox imbalance and increased oxidative damage in these subjects.

Most common biomarkers used to measure oxidative stress intensity in biological systems include lipid peroxidation products, oxidized proteins as well as oxidative modified nucleic acids. Our study demonstrated high levels of lipid (↑8-isoP, ↑4-HNE), protein (↑AOPP) and DNA (↑8-OHdG) oxidation products in CGD individuals as compared to sex- and age-matched healthy controls.

Isoprostanes are stable, sensitive and specific indicators of oxidative stress in many diseases, particularly in acute inflammatory and neurodegenerative conditions [19]. They are formed by non-enzymatic, free radical-induced peroxidation of polyunsaturated fatty acids, primarily arachidonic acid. Isoprostanes measured in our study are structurally similar to prostaglandins, considered to be important inflammatory biomarkers [20]. It has been indicated that 8-isoP increase permeability of blood vessels, stimulate chemotaxis and adherence as well as enhance the production of superoxide radicals and proinflammatory cytokines [21]. However, 8-isoP can also lead to the formation of 4-HNE, which is one of the most toxic compounds, damaging protein and DNA molecules [22]. Increased concentration of lipid peroxidation products in CGD patients may indicate destabilization of cell membranes, change in organization of the lipid bilayer and impairment of membrane lipid asymmetry. Consequently, the activity of membrane enzymes as well as the function of transporting proteins (e.g., Ca2 + -Mg2 + -ATPase) can be inhibited. The final effect of lipoperoxidation may result in impairment of cell membrane integrity and chronic inflammation [23,24].

In contrast to our study, Violi et al. found a significant reduction in the urinary excretion of isoprostanes in CGD patients compared to healthy subjects, providing evidence of reduced platelet isoprostane production in patients with hereditary deficiency of gp91phox [25]. Other studies have shown that patients with hereditary deficiency of the cytosolic subunit p47phox express more activation of NADPH oxidase and produce more isoprostanes than those with hereditary NOX2 deficiency, but less compared to healthy subjects. [26]. According to literature data, assessment of redox imbalance should take into consideration more than one parameter. The results of our study, obtained both in the whole studied group and in the groups of patients with the *CYBB* and the *NCF1* mutations, demonstrated not only an increased concentration of 8-isoP but also an increased concentration of 4-hydroxynonenal, which indicates enhanced lipid peroxidation in patients with CGD. However, the size of the study group is a limitation which does not allow for definitive conclusions to be drawn. Our study did not explore NOX2 subunit activity. Differences in the obtained results indicate a need for further studies evaluating redox imbalance in CGD, including assessment of the activity of specific NADPH subunits.

It is well established that increased activity of oxygen and nitrogen free radicals leads to the oxidation of amino acid residues, cleavage of polypeptide chains and intracellular accumulation of oxidized proteins [27]. Termination of this process leads to loss of the biological activity of proteins, which plays an important role in the pathogenesis of many diseases and cell aging [28]. AOPP are products of tyrosine oxidation in albumins, fibrinogen or lipoproteins [29]. It has been shown that AOPP stimulate pro-oxidant and proinflammatory signaling (e.g., nuclear factor kappa-light-chain-enhancer of activated B cells (NF-kB) and mitogen-activated protein kinase (MAPK)), dramatically enhancing oxygen consumption in mitochondria [28]. Moreover, intracellular accumulation of AOPP induces NOX2 activation and stimulates free radical production and cytokine storm through positive feedback [30]. AOPP also contribute to autoimmune responses, i.e., systemic lupus erythematosus (SLE), a common autoimmune manifestation in CGD individuals [31]. Lood et al. have demonstrated that the release of oxidized mitochondrial DNA with potent proinflammatory and IFN-driven properties can generate neutrophil extracellular traps (NETs) even in CGD patients with reduced or absent functional NOX2 [32]. This finding indicates that the key pathway of immune activation predisposing to autoimmunity in patients with CGD and SLE are similar [33].

Our results correlate with previous in vitro observations of intracellular oxidative stress in CGD [10]. As demonstrated by Sundqvist et al., phagocytes lacking a functional NADPH oxidase, paradoxically, generate enhanced levels of basal intracellular ROS derived from mitochondria (mtROS), leading in consequence to cytokine overproduction [10]. Furthermore, CGD patients, independently of any active chronic infection, also present inflammatory status involving all immune cell subsets. This is manifested by increased proportions of non-classic and intermediate monocytes, a proinflammatory status of mononuclear phagocytes with increased Interleukin 1β (IL-1β) and tumor necrosis factor alpha (TNFα) content, a bias of CD4+ T lymphocytes toward the T helper (Th) 17 phenotype and increased proportions of IL-17A-secreting neutrophils [34]. Based on these findings, it is assumed that phagocytic cells derived from CGD patients may have an autoinflammatory nature and can be the main source of excessive amounts of free radicals.

Redox alterations in CGD patients are confirmed by an elevated concentration of 8-oxy-2’-deoxyguanosine (↑8-OHdG), a marker of oxidative DNA damage. The effect of oxidative stress on nucleic acids is manifested by formation of numerous oxidative modifications, including damage to nitrogen bases, DNA strand breaks and generation of protein and lipid adducts. Thus, overproduction of ROS may result in gene mutations, changes in gene expression, and it may also be responsible for premature aging [35]. While the pool of intracellular proteins and lipids is periodically exchanged, nucleic acids can only be repaired by the presence of appropriate DNA repair systems [36]. Therefore, oxidative DNA damage appears to be the most dangerous for the body [37].

It might be worth emphasizing that several literature reports confirm that oxidative stress parameters are not only indicators, but also causative agents of inflammatory status or redox imbalance [38]. In view of these facts, we may suspect that the presence of oxidative tissue damage products may be the result of subclinical inflammation, but they may also exacerbate this condition (Figure 6).

To date, the overexpression of intracellular antioxidants (SOD2) in NADPH oxidase-deficient phagocytes has been confirmed [10]. To our knowledge, this is the first study demonstrating significantly enhanced serum SOD and CAT activity in children with CGD. It is well known that antioxidant enzymes (SOD, CAT and GPx) constitute the ‘first line’ of antioxidant response against ROS overproduction, upregulated mainly by proinflammatory cytokines [39]. Although we did not directly assess the ROS production rate, the observed increase in CAT and SOD activity suggests the presence of an adaptive response to the overproduction of free radicals in CGD patients. However, GPx activity did not differ significantly between the study and control groups, suggesting a high rate of ROS formation in CGD patients. It is well known that both CAT and GPx are involved in the decomposition of hydrogen peroxide, which can damage proteins, lipids, sugars and DNA. [40]. However, at physiological concentrations, this role is played by GPx, whereas in H_2_O_2_ overproduction, this role is taken over by CAT [41]. In addition, redox imbalances in favor of oxidation reactions are also confirmed by increased levels of TOS and OSI in patients with CGD. Indeed, TOS expresses the total content of oxidants in biological systems, while OSI demonstrates the balance between antioxidants and ROS production.

In our study, we did not observe any changes in uric acid or TAS concentrations in the examined plasma samples of CGD patients, although an increase in their levels is a common physiological response to oxidative stress [42,43]. TAS only reflects the current efficiency of antioxidant mechanisms, which is dependent on disease severity OR disease stage, and is regulated by many factors, including vitamins A, E, C and glutathione.

In our studies, vitamin A and E levels in CGD patients were comparable to those in healthy individuals.

To complete the picture of plasma antioxidant mechanisms, we measured not only retinol (vitamin A) and alpha-tocopherol (vitamin E), but also coenzyme Q10 concentrations, which are well known lipid-soluble free radical scavengers in the human body [44]. We observed significantly decreased coenzyme Q10 concentrations in the plasma of patients with CGD. Coenzyme Q10 is a lipophilic molecule ubiquitously present in cell membranes. It is, however, particularly abundant in the mitochondrial electron chain located in the inner mitochondrial membrane, where it transports electrons from complexes I and II onto complex III, to produce ATP [45]. Coenzyme Q10 in humans is produced mainly endogenously, with 25% of the stores obtained from dietary intake. It provides antioxidant protection of the cell membrane directly by preventing oxidative tissue damage and indirectly by regenerating other antioxidants, such as ascorbic acid or alpha-tocopherol. Coenzyme Q10 deficiency in mitochondria leads to respiratory chain dysfunctions including, i.e., bioenergetic defects and ROS overproduction [46]. Low plasma coenzyme Q10 concentrations have been observed in a variety of oxidative stress-related clinical phenotypes [47,48]. Furthermore, coenzyme Q10 modulates the function of the immune cell response. Coenzyme Q10 supplementation downregulates the alloreactivity of T cells, enhances the differentiation of Tregs in graft versus host disease (GVHD) and improves the function of T cells, leading to a reduction in infection severity [49,50].

Coenzyme Q10 deficiency is closely related to inflammation resulting from mitochondrial ROS overproduction in fibromyalgia [51]. Analogous mitochondrial pathologies have been noticed in CGD cells [10]. Based on this, we hypothesize that coenzyme Q10 deficiency also plays a role in the pathogenesis of this primary immunodeficiency disorder. In order to verify such a hypothesis, measurement of coenzyme Q10 should be performed in muscle biopsy. However, despite a lack of direct evidence, and taking into consideration the anti-inflammatory and antioxidative properties of coenzyme Q10, the results of our study indicate the need for its supplementation in patients with CGD. Effectiveness of such medication requires further research.

Unfortunately, our research has some limitations such as measurement of selected redox biomarkers and oxidative damage products and testing performed in serum and plasma samples, conditions that do not fully reflect the redox status of diseased organs. At this stage of research, it is impossible to decide whether higher amounts of free radicals result from mitochondrial ROS overproduction or from the action of proinflammatory cytokines. However, exploration of oxidant/antioxidant status in X-linked carriers of the disease and successfully transplanted CGD patients might bring further data on the analyzed biological mechanisms.

## 5. Conclusions

Our data indicate that unfavorable oxidant/antioxidant balance is a feature of chronic granulomatous disease, caused by a predisposition to develop a hyperinflammatory state due to various disturbances of the innate immune response. The enhanced enzymatic antioxidant activity and higher plasma susceptibility to oxidative stress revealed in our study may constitute a compensatory defense mechanism towards overwhelming inflammation. Since the products of ROS-mediated tissue injury influence signal transduction pathways and apoptotic cell death, we suspect that they may also contribute to the clinical progression of CGD. Nevertheless, more detailed studies are necessary to understand the relationship between oxidative stress and subclinical chronic inflammation and symptoms of autoimmunity in CGD (Figure 6). Therefore, we suggest the evaluation of mitochondrial function in blood cells of patients with CGD. Considering low coenzyme Q10 content and oxidative stress in the plasma of patients with CGD, we suggest supplementation of coenzyme Q10 in this group of patients.

Our pioneering study has clearly demonstrated systemic oxidative stress and a simultaneous compensatory plasma antioxidant response in patients with CGD. This preliminary research confirms the need for multicenter studies which would examine the impact of different types of mutations and activity of NADPH subunits on redox imbalance in patients with CGD. Increased oxidative redox parameters in CGD individuals might be of diagnostic value in evaluating the severity of subclinical inflammation in this group of patients.

## Figures and Tables

**Figure 1 jcm-09-01397-f001:**
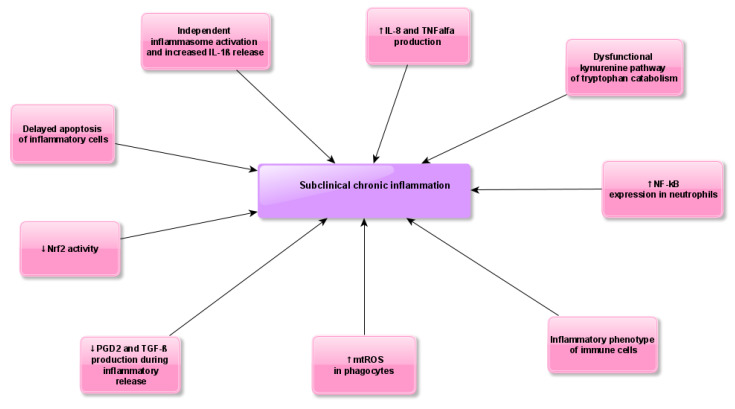
Possible mechanisms of dysregulated inflammatory response during chronic granulomatous disease (CGD). Abbreviations: IL-8, Interleukin–8; IL-1β, Interleukin–1β; Nrf2, nuclear factor erythroid 2-related factor 2; PGD2, prostaglandin D2; TGF-β, transforming growth factor beta; mtROS, mitochondrial reactive oxygen species; NF-κB, nuclear factor kappa-light-chain-enhancer of activated B cells; TNF alfa, tumor necrosis factor alfa.

**Figure 2 jcm-09-01397-f002:**
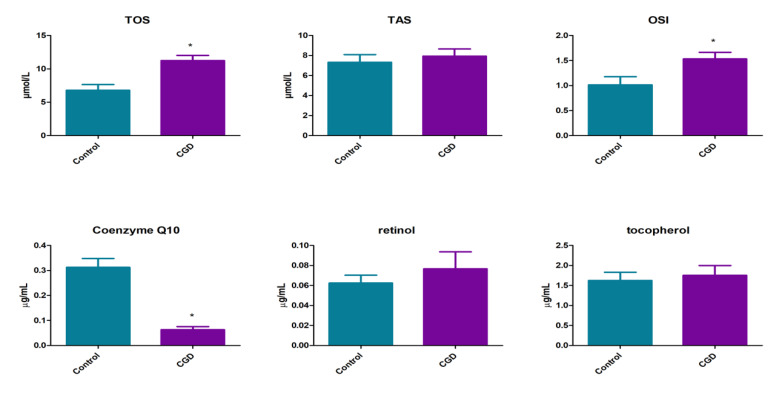
Redox status and plasma lipophilic antioxidants in patients with CGD as well as healthy controls. TOS, total oxidant status; TAS, total antioxidant status; OSI, oxidative stress index. Differences statistically significant at * *p* < 0.05.

**Figure 3 jcm-09-01397-f003:**
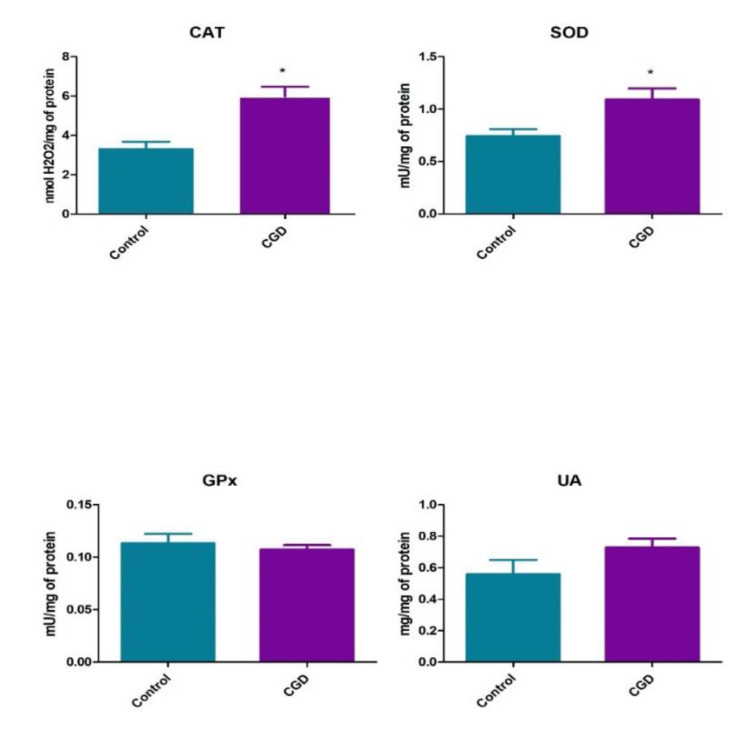
Enzymatic and non-enzymatic antioxidants in patients with CGD as well as healthy controls. CAT, catalase; SOD, superoxide dismutase; GPx, glutathione peroxidase; UA, uric acid. Differences statistically significant at * *p* < 0.05.

**Figure 4 jcm-09-01397-f004:**
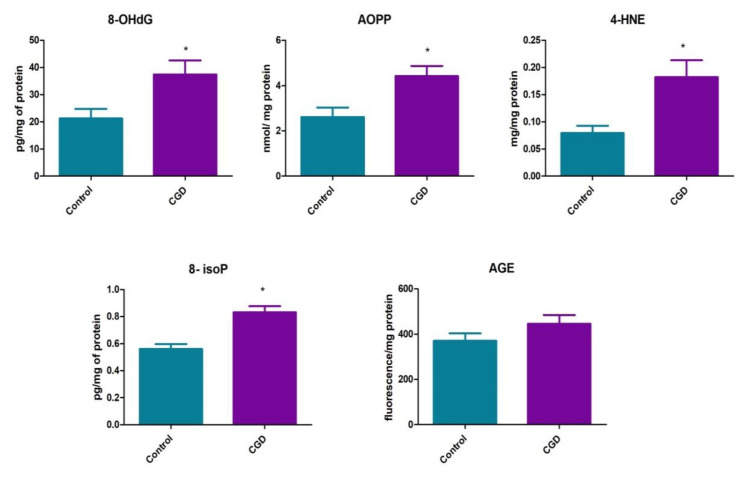
Lipid and DNA oxidation products in patients with CGD as well as healthy controls. 8-OHdG, 8-hydroxy-D-guanosine; AOPP, advanced oxidation protein products; 4-HNE, 4-hydroxynonenal; 8-isoP, 8-isoprostanes; AGE, advanced glycation end products. Differences statistically significant at * *p* < 0.05.

**Figure 5 jcm-09-01397-f005:**
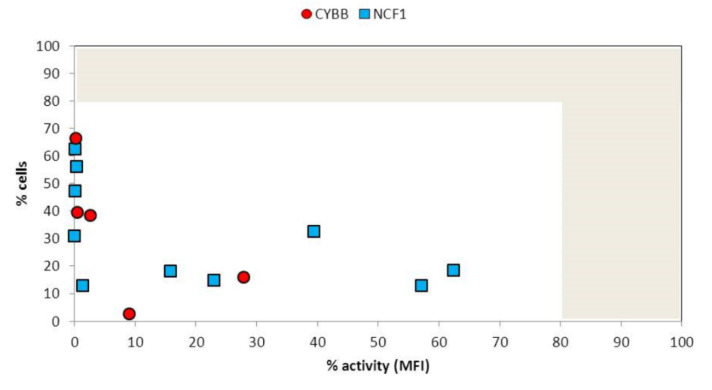
Nicotinamide adenine dinucleotide phosphate (NADPH) activity in individual patients with mutation *CYBB* (gp91phox) and *NCF1* (gp47phox). Normal range depicted as grey area (>80% of cells demonstrating >80% of activity in relation to healthy control.

**Figure 6 jcm-09-01397-f006:**
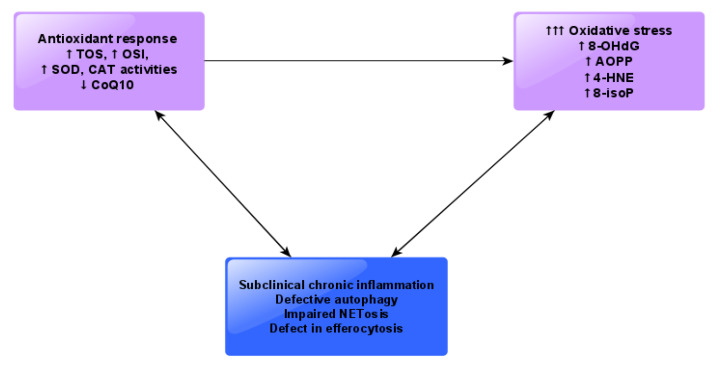
A possible link between antioxidant response, oxidative stress and subclinical chronic inflammation in patients with chronic granulomatous disease (CGD) based on our study. Abbreviations: TOS, total oxidant status; TAS, total antioxidant status; OSI, oxidative stress index; CAT, catalase; SOD, superoxide dismutase; CoQ10, Coenzyme Q10; 8-OHdG, 8-hydroxy-D-guanosine; AOPP, advanced oxidation protein products; 4-HNE, 4-hydroxynonenal; 8- isoP, 8-isoprostanes; AGE, advanced glycation end products; NET, neutrophil extracellular traps.

**Table 1 jcm-09-01397-t001:** Conditions for generating positive ions of analytes.

Compound	Precursor Ion (m/z)	Product Ion (m/z)	Collision Energy [eV]
Retinol	269.10	213.20 93.10	−12 −23
α-Tocopherol	429.30	165.10 137.05	−25 −48
Coenzyme Q10	863.60	197.15 109.10	−45 −47

**Table 2 jcm-09-01397-t002:** Basic clinical data for patients with chronic granulomatous disease (CGD).

No.	Sex (Male/Female)	Age, Years	Genetic Mutation	Age at CDG Diagnosis, Years	Clinical Manifestations Leading to Diagnosis of CGD
1	M	25	*CYBB*	6	Skin abscess
2	M	25	*CYBB*	5	Brain abscess; liver abscess; rectal abscess; pneumonitis; myocarditis; erythema nodosum
3	M	4	*CYBB*	2	Perianal abscesses
4	M	26	*CYBB*	1	Lymph nodes and skin abscesses; several with pneumonitis
5	M	28	*CYBB*	10	Pneumonia
6	M	39	*NCF1*	13	Liver abscess
7	F	24	*NCF1*	17	Liver abscesses; perianal abscesses
8	F	9	*NCF1*	7	Lymphadenitis;
9	F	13	*NCF1*	11	Recurrent pneumonia
10	F	6	*NCF1*	5	Lymphadenitis; rectal abscess
11	M	2	*NCF1*	1	Pneumonia; pulmonary aspergillosis
12	M	36	*NCF1*	34	Syphilis; skin abscesses; staphylococcal abscess of the brain and lungs
13	F	12	*NCF1*	10	Mulch pneumonia
14	M	24	*NCF1*	20	Liver abscess
15	M	26	*NCF1*	6	Liver abscess; tuberculosis of the lymph nodes

*CYBB*, gp91 phox; *NCF1*, gp47phox.

**Table 3 jcm-09-01397-t003:** Comparison of examined redox parameters in chronic granulomatous disease (CGD) patients with *CYBB* and *NCF1* mutations.

Parameter	*CYBB*, *n* = 5	*NCF1*, *n* = 10	*p*-Values
TAS, µmol/L	6.6 ± 1.8	8.5 ± 3.1	ns
TOS, µmol/L	10.6 ± 4.1	11.6 ± 2.3	ns
OSI	1.6 ± 0.7	1.5 ± 0.4	ns
CAT, nmol/100 mg of protein	6.2 ± 1.9	5.8 ± 1.9	ns
SOD, mU/mg	1.1 ± 0.2	1.1 ± 0.5	ns
GP, mU/100 mg protein	0.1± 0.01	0.1 ± 0,02	ns
UA, µg/100 mg of protein	0.8 ± 0.2	0.7 ± 0.2	ns
Coenzyme Q10, μg/mL	0.08 ± 0.04	0.05 ± 0.05	ns
Retinol, μg/mL	0.1 ± 0.09	0.08 ± 0.07	ns
Tocopherol, μg/mL	1.9 ± 1.1	1.6 ± 0.86	ns
8-OHdG, pg/mg of protein	18.0 ± 28.7	24.7 ± 19.8	ns
AOPP, nmol/mg protein	4.1 ± 1.5	4.5 ± 2	ns
4-HNE, µg/mg protein	0.17 ± 0.14	0.19 ± 0.06	ns
8-isoP, pg/mg protein	0.9 ± 0.2	0.8 ± 0.2	ns
AGE, fluorescence/mg protein	500.2 ± 168.8	443 ± 153	ns

*CYBB*, gp91 phox; *NCF1*, gp47phox. TOS, total oxidant status; TAS, total antioxidant status; OSI, oxidative stress index; CAT, catalase; SOD, superoxide dismutase; GPx, glutathione peroxidase; UA, uric acid; 8-OHdG, 8-hydroxy-D-guanosine; AOPP, advanced oxidation protein products; 4-HNE, 4-hydroxynonenal; 8-isoP, 8-isoprostanes; AGE, advanced glycation end products.

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
