# Peer review of "Systemic Redox Imbalance in Patients with Chronic Granulomatous Disease"

_jcm, 2020, doi:10.3390/jcm9051397_

Round 1

Reviewer 1 Report

The study entitled „Systemic redox imbalance in patients with chronic granulomatous disease” is very interesting, properly designed, however some minor issues need to be clarified:

  • Patients clinical data/parameters table is missing.
  • As authors state “In all cases markers of inflammation were low”, what is meant by this?
  • What antimicrobial prophylaxis was used?
  • Redox assays description in Material and methods section is not full or is placed in other chapter - Oxidative damage products?. Please provide more details or rearrange more adequately this section
  • Lines 195-197” “Both tocopherol (CGD: 76;  SD±0.94; control: 1.63; SD±0.79;  p=0.67)  and  retinol  (CGD:  0.09;  SD±0.07;  control:  0.07;  SD±0.03;  p=  0.29) plasma concentrations did not differ between the two groups (Figure 2.).” the results are depicted in Figure 1 not in Figure 2
  • Are there any correlations with patients’ clinical parameters or genetical changes?
  • The manuscript needs some better editing and grammar/punctuation check

Author Response

Reviewer 1

The study entitled 'Systemic redox imbalance in patients with chronic granulomatous disease' is very interesting, properly designed, however some minor issues need to be clarified:

  • Patients clinical data/parameters table is missing.

Thank you for your suggestion. We have prepared Table 1 with clinical data of patients with CGD. During the preparation of the manuscript, incorrect data on CGD patients was entered. We have corrected the data in the manuscript. The study group consisted of 15 patients including 10 patients with the NCF1 mutation and 5 patients with the CYBB mutation.

Table 2. Basic clinical data patients with chronic granulomatous disease.

No.

Sex (Male/Female)

Age, years

Genetic mutations

Age at CDG diagnosis, years

Clinical manifestations
leading to diagnosis of CGD

1

M

25

CYBB

6

skin abscess,

2

M

25

CYBB

5

brain abscess; liver abscess; rectal abscess; pneumonitis; myocarditis; erythema nodulare

3

M

4

CYBB

2

perianal abscesses

4

M

26

CYBB

1

lymph nodes and skin abscesses; several pneumonitis

5

M

28

CYBB

10

pneumonia

6

M

39

NCF1

13

liver abscess

7

F

24

NCF1

17

liver abscesses, perianal abscesses

8

F

9

NCF1

7

lymphadenitis;

9

F

13

NCF1

11

recurrent pneumonia

10

F

6

NCF1

5

lymphadenitis, rectal abscess

11

M

2

NCF1

1

pneumonia, pulmonary apergillosis

12

M

36

NCF1

34

syphilis; skin abscesses; staphylococcal abscess of the brain and lungs

13

F

12

NCF1

10

mulch pneumonia

14

M

24

NCF1

20

liver abscess

15

 M

26

NCF1

6

liver abscess; tuberculosis of the lymph nodes

CYBB, gp9 phox, NCF1, gp47phox

  • As authors state “In all cases markers of inflammation were low”, what is meant by this?-

We have corrected this sentence in Material and methods. In all patients with CGD, markers of inflammation (C-reactive protein CRP, leukocytosis, erythrocyte sedimentation rate, ESR) were negative.

  • What antimicrobial prophylaxis was used?

We have added this information to the Material section. At the time of study commencement, all CGD patients were receiving itraconazole, trimethoprim and sulfamethoxazole.

  • Redox assays description in Material and methods section is not full or is placed in other chapter - Oxidative damage products? Please provide more details or rearrange more adequately this section

We have rearranged this section. We have added subpoints in Redox assays: a/ Antioxidant barrier and b/ Oxidative damage products of proteins, lipids and DNA

  • Lines 195-197” “Both tocopherol (CGD: 76 SD±0.94; control: 1.63; SD±0.79; p=0.67) and retinol (CGD: 0.09; SD±0.07; control: 0.07; SD±0.03; p= 0.29) plasma concentrations did not differ between the two groups (Figure 2.).” the results are depicted in Figure 1 not in Figure 2

Thank you for pointing this out. Fig. 2 has been changed to Fig.1.

  • The manuscript needs some better editing and grammar/punctuation check

The manuscript has been checked and corrected.

  1. Are there any correlations with patients’ clinical parameters or genetical changes?

We did not find correlations between clinical parameters or genetic changes.

  • One factor that may determine implications for the results is the fact that the percentages of the various mutations present in the general population have not been maintained in the study population (CYBB is present in 60-70% of cases in the affected population), with patients with a NCF1 mutation prevailing in the study population???

We have added Table 3. We present a comparison of selected redox parameters between patients with the CYBB and the NCF 1 mutation. There were no differences in the assessed parameters of redox status between patients with the CYBB and the NCF1 mutation.

  1. There are more women than men, which does not correspond to the general population.

Fifteen Caucasian patients with CGD (five females and ten males, were recruited for this study. The mutations of NCF1 gene were confirmed in ten patients, mutations in CYBB gene ‒ in five patients. We have prepared Table 1 with clinical data of patients with CGD.

  • The exclusion criteria should include or have ruled out in patients: cystic fibrosis, inflammatory bowel disease, hyper IgE syndrome, allergic broncho-pulmonary aspergillosis, glutathione synthetase or myeloperoxidase deficiency, and secondary hemophagocytic lymphohistiocytosis.

We have added this information and change line 88 -98.
All patients included in the study were found to be in good health and were receiving itraconazole, trimethoprim and sulfamethoxazole at the time of enrollment. No side effects of the drugs used were observed Apart from CGD, none of the patients demonstrated any other pathologies or exclusion criteria, i.e. metabolic diseases (obesity, diabetes), hypertension, heart, liver, kidney or lung diseases (cystits fibrosis, allergic broncho-pulmonary aspergillosis), HIV infection, cancer, hyper IgE syndrome, glutathione synthetase or myeloperoxidase deficiency and secondary hemophagocytic lymphohistiocytosis. Based on histopathological examination, six patients with CGD showed mild IBD, but without clinical symptoms and not requiring treatment. In all patients with CGD, markers of inflammation (C-reactive protein CRP, leukocytosis, erythrocyte sedimentation rate, ESR ) were negative. The type and degree of compliance with prophylaxis is not described, which may have an impact on redox activity.

We've added this information to the Material section / line No 87

Best regards,

Reviewer 2 Report

One factor that may determine implications for the results is the fact that the percentages of the various mutations present in the general population have not been maintained in the study population (CYBB is present in 60-70% of cases in the affected population), with patients with a NCF1 mutation prevailing in the study population.

There are more women than men, which does not correspond to the general population.

The exclusion criteria should include or have ruled out in patients: cystic fibrosis, inflammatory bowel disease, hyper IgE syndrome, allergic broncho-pulmonary aspergillosis, glutathione synthetase or myeloperoxidase deficiency, and secondary hemophagocytic lymphohistiocytosis.

The type and degree of compliance with prophylaxis is not described, which may have an impact on redox activity.

Author Response

  • The authors evaluated the redox status of 15 patients with CGD, 9 with mutations in the NCF1 gene (p47phox) and six with mutations in the CYBB gene (gp91phox). Mutations in these genes can result in complete abolition or inadequate protein production, so it is important to know what the patient's phenotype was in terms of neutrophil oxidase activity (DHR stimulation index) and this should be reported in the text. The residual activity of NADPH oxidase is known to have a strong impact on the clinical course and outcome of these patients, (M Y Koker J Allergy Clin Immunol 2013 Nov 132)

  1.  

We have added this information to the Material section in graphic form. NADPH activity in patients with chronic granulomatous disease measured as % of granulocytes demonstrating change in dihydrorhodamine fluorescence after stimulation with phorbol myristate presented in Fig. 4.

Fig. 4. NADPH activity of individual patients with CYBB and NCF1 mutations.

Best regards,

Reviewer 3 Report

1) The authors evaluated the redox status of 15 patients with CGD, 9 with mutations in the NCF1 gene (p47phox) and six with mutations in the CYBB gene (gp91phox). Mutations in these genes can result in complete abolition or inadequate protein production, so it is important to know what the patient's phenotype was in terms of neutrophil oxidase activity (DHR stimulation index) and this should be reported in the text.  The residual activity of NADPH oxidase is known to have a strong impact on the clinical course and outcome of these patients, ( M Y Koker J Allergy Clin Immunol 2013 Nov 132 ) and this could be a bias for  the redox status analysis.

2)  The authors demonstrated high levels of plasma 8-isoprostane in CGD individuals. In contrast, Violi et al (Violi F. Circulation 2009, 20 October 2009) found a significant reduction in the urinary excretion of isoprostanes in X-CGD patients compared to healthy subjects, providing evidence of reduced platelet isoprostane production from patients with hereditary deficit of gp91phox.  
Other studies have shown  that patients with hereditary deficiency of the cytosolic sub-unit p47phox express more activation of NADPH oxidase and produce more isoprostanes than those with NOX2 hereditary deficiency but lower compared to healthy subjects. Moreover patients with  deficiency of NOX2, elicits deeper reduction of oxidant species compared to the deficiency of a cytosolic sub-unit such as p47phox (Carnevale R. Br.J Haematol 2018 Feb 180 , J Am Heart Assoc2014 Jun 27). These data should be discussed and the results  described separately for the two patient groups  ( with gp91phox and p47phox deficiency ).

Author Response

  • The authors demonstrated high levels of plasma 8-isoprostane in CGD individuals.

The authors demonstrated high levels of plasma 8-isoprostane in CGD individuals. In contrast, Violi et al (Violi F. Circulation 2009, 20 October 2009) found a significant reduction in the urinary excretion of isoprostanes in X-CGD patients compared to healthy subjects, providing evidence of reduced platelet isoprostane production from patients with hereditary deficit of gp91phox.

Other studies have shown that patients with hereditary deficiency of the cytosolic sub-unit p47phox express more activation of NADPH oxidase and produce more isoprostanes than those with NOX2 hereditary deficiency but lower compared to healthy subjects. Moreover patients with deficiency of NOX2, elicits deeper reduction of oxidant species compared to the deficiency of a cytosolic sub-unit such as p47phox (Carnevale R. Br.J Haematol 2018 Feb 180, J Am Heart Assoc2014 Jun 27). These data should be discussed and the results described separately for the two patient groups (with gp91phox and p47phox deficiency). Other studies have shown that patients with hereditary deficiency of the cytosolic sub-unit p47phox express more activation of NADPH oxidase and produce more isoprostanes than those with NOX2 hereditary deficiency but lower compared to healthy subjects. Moreover, patients with deficiency of NOX2, elicits deeper reduction of oxidant species compared to the deficiency of a cytosolic sub-unit such as p47phox (26). Other studies have shown that patients with hereditary deficiency of the cytosolic sub-unit p47phox

We added this information in section Discussion line No 285-304. In contrast to our study, Violi et al. found a significant reduction in the urinary excretion of isoprostanes in X-CGD patients compared to healthy subjects, providing evidence of reduced platelet isoprostane production in patients with hereditary deficiency of gp91phox [25]. However, our study demonstrated increased concentrations of both 8-isoprostanes and 4-hydroxynoneal protein adducts, which confirms the process of lipid oxidation in patients with CGD. The contradictory results indicate a need for further studies evaluating redox imbalance in CGD, including the activity of specific NADPH enzymatic subunits. The present study did not involve the assessment of NOX2 subunit NADPH activity.

Other studies have shown that patients with hereditary deficiency of the cytosolic sub-unit p47phox produce more isoprostanes than those with hereditary NOX2 deficiency but lower compared to healthy subjects. [26]. According to literature data, the assessment of redox inbalance should take into consideration more than one parameter of these processes. The results of our study, both in the whole studied group and in the group of patients with the CYBB and NCF 1 mutations, demonstrated not only increased concentration of 8-isoprostanes but also increased concentration of 4-hydroxynoneal, which indicates enhanced lipid oxidation in patients with CGD. Significantly decreased concentration of Coenzyme Q10 and elevated markers of lipid peroxidation (8-isoprostanes and 4 HNE) are common to the two mutations. However, the size of the study group is a limitation which does not allow for definitive conclusions to be drawn.

  • These data should be discussed and the results described separately for the two patient groups (with gp91phox and p47phox deficiency).

Our study confirmed redox disturbances and increased oxidative damage in CGD patients as well as indicates the need to compare redox imbalance depending on the type of mutation and NADPH activity.

However, exploration of oxidant/antioxidant status in X linked carriers of the disease and successfully transplanted CGD patients might bring further data on the analyzed biological mechanisms.

We have added:

We consider our research to be preliminary. It confirms the need for multi-centre studies examining the impact of different types of mutations and activity of specific NADPH subunits on redox imbalance in patients with CGD. Increased oxidative redox parameters in CGD individuals might by of diagnostic value in evaluating the severity of subclinical inflammation in this group of patients. Line No. 420 - 422.

Table 3. Comparison of examined redox parameters in CGD patients with CYBB and NCF1 mutations.

Parameter

CYBB, n = 5

NCF1, n = 10

p Values

TAS, µmol/L

6.6 ± 1.8

8.5 ± 3.1

ns

TOS, µmol/L

10.6 ± 4.1

11.6 ± 2.3

ns

OSI

1.6 ± 0.7

1.5 ± 0.4

ns

CAT, nmol/ mg of protein

6.2 ± 1.9

5.8 ± 1.9

ns

SOD, mU/mg

1.1 ± 0.2

1.1 ± 0.5

ns

GP, mU/ mg protein

0.1± 0.01

0.1 ± 0,02

ns

UA, µg/ mg of protein

0.8 ± 0.2

0.7 ± 0.2

ns

Coenzyme Q10, μg/mL

0.08 ± 0.04

0.05 ± 0.05

ns

Retinol, μg/mL

0.1 ± 0.09

0.08 ± 0.07

ns

Tocopherol, μg/mL

1.9 ± 1.1

1.6 ± 0.86

ns

8-OHdG, pg/mg of protein

18. ± 28.7

24.7 ± 19.8

ns

AOPP, nmol/ mg protein

4.1 ± 1.5

4.5 ± 2

ns

4-HNE, µg/mg protein

0.17 ± 0.14

0.19 ± 0.06

ns

8-isop, pg/mg protein

0.9 ± 0.2

0.8 ± 0.2

ns

AGE, fluorescence/mg protein

500.2 ± 168.8

443 ± 153

ns

CYBB, gp91 phox, NCF1, gp47phox

Round 2

Reviewer 1 Report

I am satisfied with authors' response. Accept in the present form.